# Peer review of "A novel facial expression recognition framework using deep learning based dynamic cross-domain dual attention network"

_PeerJ Computer Science, doi:10.7717/peerj-cs.2866_

## Round 0.1 · original submission · Major Revisions

Dear Authors,


Reviewers have now commented on your article. A substantial revision is required to substantiate the validation of the results presented and in comparison to those of the state of the art. Minor refinements in statistical analysis and reporting clarity are required.Hhyperparameter selection and training efficiency details should be clarified. Minor English grammar and writing style errors should also be corrected.

We do encourage you to address the concerns and criticisms of the reviewers with respect to reporting, experimental design, and validity of the findings and resubmit your article once you have updated it accordingly.

Best wishes,

**Language Note:** The Academic Editor has identified that the English language must be improved. PeerJ can provide language editing services - please contact us at [email protected] for pricing (be sure to provide your manuscript number and title). Alternatively, you should make your own arrangements to improve the language quality and provide details in your response letter. – PeerJ Staff

·

Basic reporting

- The manuscript is written in clear and professional English, but there are minor grammatical inconsistencies. Proofreading could further enhance readability.
- The introduction provides sufficient background on facial expression recognition (FER) challenges and the motivation for the study. However, the related work section could better highlight how this approach significantly differs from previous studies.
- The manuscript is well-structured, adhering to standard scientific conventions, with clearly labeled sections and subsections.
- Figures are high quality and relevant, providing valuable insights into the methodology and results. However, the captions for Figures 4 and 5 could be more descriptive.
- Raw data and references are provided, meeting PeerJ’s reporting standards.
- More depth Literature Survey is required.

Experimental design

- The research is within the scope of the journal and presents original work on FER using deep learning and domain adaptation techniques.
- The research question is well-defined and addresses the problem of domain shifts in FER. The authors clearly explain how their model improves upon existing methods.
- The methodology is generally rigorous, but some details regarding hyperparameter tuning and the training process could be expanded for reproducibility. For instance, the selection criteria for the hyperparameters α and β in Equation 4 are not explicitly justified.
- The use of multiple datasets (RAF-DB, FER-PLUS, AffectNet, etc.) strengthens the validity of the study, but the authors should clarify whether any dataset augmentation techniques were used.

Validity of the findings

- The study presents strong experimental results, demonstrating the model's robustness across multiple datasets.
- The statistical analysis appears sound, but additional details about the significance of the improvements over baseline models would strengthen the conclusions. A statistical test (e.g., t-test) comparing model performance across datasets could provide more rigor.
- The conclusions align with the results but should be slightly refined to emphasize practical applications and limitations. While the paper claims that the model "consistently outperforms state-of-the-art techniques," a discussion on potential failure cases or computational constraints would provide a more balanced perspective.
- Cited papers are not having results mentioned in the table. Cross check once again.
- Results mentioned in the table and Confusion Matrix are not justifying.
- Confusion Matrix clearly shows for 1-2 classes are not at all classified. Which is major drawback of your method.

Additional comments

- Strengths:
- The study tackles an important challenge in FER and offers a novel approach that integrates both global and local adversarial learning.
- The evaluation on multiple datasets makes the findings more generalizable.
- The proposed semantic-aware pseudo-labeling method is an interesting contribution.
- Weaknesses:
- While the dual-attention mechanism is well-explained, more details on the training efficiency and computational overhead would be beneficial.
- The authors should discuss how the model could perform in real-time applications and whether additional computational resources are needed compared to conventional FER models.
- The manuscript should be revised for minor grammatical errors and clarity in certain sections.
- Cite the more papers to strengthen the literature survey.

·

Basic reporting

1. I have conducted a comprehensive review of the paper and found that the English language used is professional, precise, and unambiguous.

2. The literature review is well-developed, incorporating an adequate number of relevant references.

3. The paper is well-structured.

4. Section 3 has not used any references, not even one. If none has been used, please give a background for equations 1 to 4.

5. Equations 1 to 4 need mathematical proof.

Experimental design

1. What kind of optimization is used for equation 4?

2. Explain the integration steps between the proposed algorithm with ResNet50 and MobileNet-V2—for example, line 407.
3. Some of the results show low mean accuracy (lines 356 to 400)

4. Plot a block diagram showing the process of pairing with a dual attention mechanism.

5. I suggest adding a new section about the background theory for the proposed model.

6. The paper does not provide sufficient details regarding the implementation settings, for example the programming language used, deep learning framework, number of epochs, batch size, learning rate, optimizer, and other hyperparameters. Including this information would enhance the reproducibility of the study and allow other researchers to validate the results. It would be beneficial to add a section detailing the experimental setup, specifying the computational resources used (such as GPU type and memory), as well as the training configurations.

Validity of the findings

1. To the best of my knowledge, the paper presents a novel contribution to the field. The study’s contributions are clearly articulated and effectively contextualized.

2. Additionally, the conclusions are well-stated, directly addressing the research questions posed in the study and substantiated by the presented results.

---

## Round 0.2 · Major Revisions

Dear Authors

Please clearly address the concerns and criticisms of Reviewer 1 and resubmit your paper once you have updated it. Reviewer has asked you to provide specific references. You are welcome to add them if you think they are useful and relevant. However, you are under no obligation to include them, and if you do not, it will not affect my decision.

Best wishes,

·

Basic reporting

The manuscript is generally well-structured and follows a logical flow. However, there are inconsistencies in Table 1 regarding cross-domain accuracy using FERPlus and ResNet50 on AffectNet, EXP-W, SFEW2.0, and JAFFE datasets. The provided results do not match the cited references, which raises concerns about the accuracy of the data.

The literature review needs significant improvement. The cited references are outdated, and more recent works should be included to reflect current advancements in the field. It is recommended that the authors refer to the following review papers for a better literature survey:

Borgalli, R. A., & Surve, S. (2022). "Review on learning framework for facial expression recognition." The Imaging Science Journal, 70(7), 483–521. https://doi.org/10.1080/13682199.2023.2172526

Borgalli, R.A., & Surve, S. (2025). "A Hybrid Optimized Learning Framework for Compound Facial Emotion Recognition." In Proceedings of Fifth International Conference on Computing, Communications, and Cyber-Security (IC4S 2023). Lecture Notes in Networks and Systems, vol 1128, Springer, Singapore. https://doi.org/10.1007/978-981-97-7371-8_35

Experimental design

The experimental setup is described clearly, but the justification for selecting FERPlus as the source dataset and ResNet50 as the backbone network is not well-elaborated. The authors should provide a rationale for these choices and compare their model’s performance with other architectures such as Vision Transformers or EfficientNet.

It is unclear whether the dataset splits follow standard benchmarks for training, validation, and testing. The authors should explicitly mention the dataset partitioning strategy to ensure reproducibility.

Validity of the findings

The reported results lack proper statistical validation. Confidence intervals, standard deviations, or significance tests should be included to strengthen the findings.

The claims regarding improved cross-domain accuracy should be supported by comparative results with baseline methods. Additionally, visual examples of misclassified expressions can provide deeper insights into the model’s limitations.

Additional comments

The manuscript needs better clarity in explaining key terms and methodologies. Adding a brief background on facial expression recognition (FER) and recent deep learning advancements in this domain would help readers unfamiliar with the field.
The conclusion should summarize key findings and discuss future directions more explicitly.

---

## Round 0.3 · accepted · Accept

Dear Authors,

One of the invited reviewers did not respond to the invitation to provide a review for the revised paper.. However, one reviewer has accepted the paper. I have also conducted my own assessment of the revision, and I am satisfied with the current version.

Best wishes,

·

Basic reporting

As this version is the revised one of the manuscript, were 4 insightful comments and suggestions have been forwarded to the authors for constructive feedback. All the insightful comments and suggestions have been constructive to enhance the quality of the manuscript. All the responses and feedback from the authors were satisfactory and aligned with the reviewer's expectations.

Moreover, the authors have corrected all the required grammatical errors, and typographical issues, and improved the overall language quality. The manuscript is generally well-structured and follows a logical flow. Tables were corrected and relevant references have been added to the paper.

Experimental design

Suggestions from the reviewers regarding the justification for selecting FERPlus as the source dataset and ResNet-50 as the backbone network. Basically, FERPlus dataset was chosen due to its extensive annotation of facial expressions, which provides a more granular understanding of subtle emotional variations compared to traditional FER datasets whereas FERPlus offered a more robust foundation for training and evaluation.

The authors further evaluated different statistical parameters including confidence intervals, standard deviations, and significance tests that ultimately show the reliability, and authenticity of the performance of the proposed model

Validity of the findings

The authors have updated the description of the conclusion section by including the performance figures and the significance of the proposed solution.